# Apple CRISPR-Cas9—A Recipe for Successful Targeting of *AGAMOUS*-like Genes in Domestic Apple

**DOI:** 10.3390/plants12213693

**Published:** 2023-10-26

**Authors:** Seth Jacobson, Natalie Bondarchuk, Thy Anh Nguyen, Allison Canada, Logan McCord, Timothy S. Artlip, Philipp Welser, Amy L. Klocko

**Affiliations:** 1Department of Biology, University of Colorado Colorado Springs, Colorado Springs, CO 80918, USA; 2U.S. Department of Agriculture, Agricultural Research Service (USDA-ARS), The Appalachian Fruit Research Station, 2217 Wiltshire Road, Kearneysville, WV 25430, USA; philipp.welser@usda.gov

**Keywords:** CRISPR-Cas9, gene editing, *AGAMOUS*, *MADS*, domestic apple, biotechnology

## Abstract

Fruit trees and other fruiting hardwood perennials are economically valuable, and there is interest in developing improved varieties. Both conventional breeding and biotechnology approaches are being utilized towards the goal of developing advanced cultivars. Increased knowledge of the effectiveness and efficiency of biotechnology approaches can help guide use of the CRISPR gene-editing technology. Here, we examined CRISPR-Cas9-directed genome editing in the valuable commodity fruit tree *Malus* x *domestica* (domestic apple). We transformed two cultivars with dual CRISPR-Cas9 constructs designed to target two *AGAMOUS*-like genes simultaneously. The main goal was to determine the effectiveness of this approach for achieving target gene changes. We obtained 6 Cas9 control and 38 independent CRISPR-Cas9 events. Of the 38 CRISPR-Cas9 events, 34 (89%) had gene edits and 14 (37%) showed changes to all alleles of both target genes. The most common change was large deletions, which were present in 59% of all changed alleles, followed by small deletions (21%), small insertions (12%), and a combination of small insertions and deletions (8%). Overall, a high rate of successful gene alterations was found. Many of these changes are predicted to cause frameshifts and alterations to the predicted peptides. Future work will include monitoring the floral development and floral form.

## 1. Introduction

Fruit crops are important sources of food and nutrition for humans. A 2022 estimate of global fruit production was 890 million tons worldwide, with the United States alone accounting for 470 million tons [1]. Major fruit crops include *Solanum lycopersicum* (tomato), *Musa* spp. (bananas and plantains), *Citrullus lanatus* (watermelon), *Malus* x *domestica* (domestic apple), and *Citrus* spp. (citrus) [1]. Given increased consumer demand, both conventional breeding and genetic engineering (GE) approaches are being used to obtain fruit cultivars with altered traits to better address some of the needs of these crops and to meet consumer demand [2,3,4]. Desired changes include improved disease resistance, abiotic stress tolerance, increased nutritional value, and reduced post-harvest waste [5,6,7,8]. Both conventional and GE approaches can sometimes lead to the same desired outcome. Such was the case for red-fleshed domestic apples, obtained either from breeding with red-fleshed cultivars [9] or from overexpression of the apple transcription factor *MYB10* for the anthocyanin biosynthesis pathway [10]. Similarly, purple-fruited tomatoes, also with higher anthocyanin content, have been achieved either from trait introgression from a wild relative [11] or by creating a transgenic variety with genes from *Antirrhinum* (snapdragon), a non-breeding-compatible species [12]. Sometimes, a novel fruit coloration trait achieved via genetic engineering is distinct enough for marketing, as is the case of the pink-fruited *Ananas comosus* (pineapple) recently released by Del Monte Fresh Produce [13].

CRISPR-Cas9 genome editing has been successfully applied to a wide range of agricultural, ornamental, and horticultural crops [14,15].This approach shows great promise for achieving desired changes in fruit crops of interest. Numerous proof-of-concept studies have shown that it is possible to achieve stable CRISPR-Cas9 genome edits in various fruit and nut species, including apple, banana, *Vaccinium corymbosum* (blueberry), *Theobroma cacao* (cacao), *Castanea sativa* (sweet chestnut), citrus, *Coffea canephora* (coffee), *Vitis vinifera* (grape), *Pyrus communis* (pear), *Actinidia deliciosa* (kiwi), and *Punica granatum* (pomegranate) (Table 1, [16,17,18,19,20,21,22,23,24,25,26,27,28,29,30,31]). There is interest in developing approaches in many other fruit species, including tropical fruits, but progress is hampered by a need for improved genetic resources, micropropagation, gene transfer abilities, and more [32]. A recent study in apple found that editing the *SQUAMOSA PROMOTER BINDING PROTEIN-LIKE 6* (*MdSPL6*) gene led to increased shoot formation in tissue culture; perhaps this approach could be applied to challenging species or cultivars [33]. For most proof-of-concept studies, the gene *PHYTOENE DESATURASE* (*PDS*) was targeted, as the inactivation of gene function leads to the development of albino cells, which can be readily visualized during the regeneration process. However, alterations in most genes of interest do not lead to phenotype changes that can be visually assessed in tissue culture. Therefore, some initial studies of CRISPR-Cas9 gene editing of non-*PDS* targets assessed genetic changes to the targeted gene or genes (Table 1).

Many applied studies of CRISPR-Cas9 gene editing in fruit crops are aimed at achieving improved disease resistance. This biotic stress is a major cause of reductions in crop yield [34,35,36]. Improved resistance to both viral and fungal diseases has been achieved in banana, citrus, grape, and cacao [37,38,39,40,41,42,43,44]. Other traits of interest are related to accelerated breeding and crop production. Early-flowering varieties of apple and pear and were achieved via editing of *TFL1*, which should shorten the breeding cycle [16]. Bananas with a semi-dwarf form were obtained by simultaneous editing of sites in a gibberellin pathway gene present in multiple copies, which should both assist in ease of harvesting and reduce losses from lodging [45,46]. Bananas with longer fruit shelf life were obtained from edits to a gene in the ethylene synthesis pathway [47]. Other crop-specific changes have been achieved as well, such as grapes with decreased tartaric acid content and apples with lowered foliar phloridzin [48,49].

**Table 1 plants-12-03693-t001:** Examples of CRISPR-Cas9 gene editing in perennial fruit and nut species.

Crop Species	Gene(s) Targeted	Outcome	References
Proof-of-concept studies			
apple	*PDS*	Albino and sectored explants	[22]
apple	*PDS*	Albino and sectored explants	[16]
apple	*PDS*	Target gene changes	[23]
apple	*DIPM-1*, *DIPM-2*, *DIPM-4*	Target gene changes	[19]
banana	*PDS1*, *PDS2*	Albino and variegated explants	[18]
banana	*PDS*	Albino and pale green explants	[20]
blueberry	*PDS*	Albino and sectored explants	[26]
chestnut	*PDS*	Albino explants	[24]
citrus—Carrizo citrange	*PDS*	Albino and sectored explants	[28]
citrus—grapefruit	*PDS*	Target gene changes	[17]
citrus—mini-citrus	*PDS*	Albino and sectored explants	[29]
citrus—mini-citrus	*CCD4*	Target gene changes	[29]
coffee	*PDS*	Albino explants	[30]
grape	*IdnDH*	Target-site changes	[23]
grape	*PDS*	Variegated explants	[21]
grape	*PDS*	Albino explants	[25]
grape	*MLO-7*	Target gene changes	[19]
kiwi	*PDS*	Albino explants	[27]
pomegranate	*PgUGTs*	Target-site changes	[31]
Applied work—improved disease resistance			
apple	*MdDIPM4*	Resistance to fire blight	[40]
banana	*eBSV* (viral gene)	Resistance to banana streak virus	[41]
cacao	*TcNPR3*	Resistance to *Phytophthora tropicalis*	[37]
citrus—grapefruit	*CsLOB1*	Resistance to citrus canker	[38]
citrus—Wanjincheng orange	*CsLOB1* (promoter)	Resistance to citrus canker	[39]
citrus—Wanjincheng orange	*CsWRKY22*	Resistance to citrus canker	[43]
grape	*VvMLO3*	Resistance to powdery mildew	[42]
grape	*VvWRKY52*	Resistance to *Botrytis cinerea*	[44]
Applied work—shortened juvenile period			
apple	*TFL1*	Early flowering	[16]
pear	*TFL1*	Early flowering	[16]
Applied work—plant form			
banana	*MaGA20ox2*	Semi-dwarf size	[46]
Applied work—fruit shelf life			
banana	*MaACO1*	Increased shelf life	[47]
Applied work—altered metabolite content			
apple	*MdPGT1*	Reduced phloridzin levels	[49]
grape	*IdnDH*	Reduced tartaric acid levels	[48]
Applied work—increased shoot regeneration			
apple	*MdSPL6*	Increased shoot regeneration	[33]

Apple is a valuable fruit crop, with yearly global apple production in 2021 reaching 9.4 million tons [1]. It would be advantageous to improve resistance to key biotic stresses, including fire blight and apple scab, as these factors severely impact tree health and decrease marketable fruit yield. There are resistant wild relatives available for trait introgression into domestic apple [50,51]. However, the traditional breeding approach of using compatible relatives to add traits is challenging in apple, due in part to the long juvenile period, as well as the lack of self-compatibility and the genetic heterozygosity of apple cultivars in general [52]. Apple is amenable to a variety of biotechnology methods, and these approaches have been used to achieve valuable traits. Early-flowering apple, which could be used to accelerate the breeding cycle, has been obtained through diverse approaches, including virus-induced expression of *Arabidopsis thaliana* (*Arabidopsis*) *FLOWERING LOCUS T* (*AtFT*) [53], overexpression of *Betula pendula* Roth (silver birch) *BpMADS4* [54], stable overexpression of an apple *FT* homolog *MdFT1* [55], RNAi of apple *TERMINAL FLOWER 1* (*TFL1*) homologs [56], and CRISPR targeting of apple *TFL1* [16]. The early-flowering *BpMADS4* trees are being used in accelerated breeding programs for high-value trait introgression [57,58]. Improved rooting and reduced tree size occurred with expression of the gene *ROOTING LOCUS B* (*rolB*) from *Agrobacteria rhizogenes* in rootstocks grafted to non-transgenic scions [59]. Overexpression of the *Lc* gene from *Zea maize* (corn) led to improved resistance to both apple scab and fire blight, but tree form was altered in commercially unacceptable ways [60]. Improved resistance to fire blight has also been achieved by CRISPR-mediated genome editing of the *MdDIPM4* gene in apple; however, this study did not report the overall vegetative form [40]. There is a precedent for applying these studies to apple production, as one variety of genetically engineered apple with non-browning fruits obtained from RNAi suppression of *POLYPHENOL OXIDASE* (*PPO*) has been commercialized [61]. Therefore, additional knowledge of the efficacy of biotechnology approaches in apple could be applicable to cultivar improvement and commercial production. 

A goal of our work was to determine the efficiency and efficacy of dual-gene editing with CRISPR-Cas9 in domestic apple. Prior work with CRISPR gene targeting in apple demonstrated that individual genes can be successfully targeted and edited with gene-specific single-guide RNAs (sgRNAs) [16,19,22,23]. Apple has a diploid genome; however, there is evidence of a historical whole-genome duplication event, and many gene families are greatly duplicated [62]. It would be advantageous to perform editing of two or more genes at the same time, making use of conserved gene regions. For this project, we used CRISPR-Cas9 to target two apple homologs of the key floral development gene *AGAMOUS* (*AG*). We previously identified *MdMADS15* (hereafter *MADS15*) and *MdMADS221* (hereafter *MADS221*) as close relatives of *AG* [63]. Our prior work utilizing RNA interference (RNAi) to suppress *MADS15* and *MADS221* in the ‘Galaxy’ cultivar reduced both male and female fertility, reduced floral determinacy, and led to extra petals through the homeotic conversion of stamens to petals [63]. An applied goal of this prior study was to reduce fertility and achieve decreased invasive potential while retaining blooms. However, even strongly suppressed RNAi events still showed some target gene expression, as well as reduced expression of the non-target gene *MdMADS14.* Additionally, the ‘Galaxy’ cultivar previously used for RNAi has the unusual apple trait of fruit set in the absence of pollination, which was an unexpected outcome. Our current CRISPR-Cas9 targeting project utilized a more precise gene-targeting method and two cultivars of domestic apple, ‘Royal Gala’ and ‘M.26’, that should only set fruit after successful pollination and seed set. This work will add to the knowledge of the targeting efficiency of editing two genes simultaneously in a tree fruit crop and could be applied to reduce the invasive potential of flowering ornamental fruiting species such as *Pyrus calleryana* (Bradford pear), a tree initially popular as an ornamental planting and a source of breeding for fire-blight resistance but now recognized as an invasive species in several states [64]. 

## 2. Results

### 2.1. Analysis of Transformation Efficacy

Prior analysis of the *MADS* family from domestic apple indicated that two genes, *MADS15* and *MADS221*, appear to be homologous to *AG* [63], with two alleles per gene. As CRISPR-Cas9 gene targeting relies on precise matches between sgRNAs and target sites, we amplified and sequenced portions of the *MADS15* and *MADS221* genes from both ‘Royal Gala’ and ‘M.26’ cultivars. Primers were designed based on the published domestic apple genome [62].

We designed a set of sgRNAs to target our two genes simultaneously (Appendix A). The sgRNA sites were located in the first exon of each gene to hopefully disrupt protein coding (Figure 1). We created four different sgRNA constructs, each with a different pairing of sgRNAs (pairings and sgRNA sequences are shown in Appendix A). We used two sgRNAs in each gene-editing construct to increase target gene editing frequency and potentially create larger deletions (Figure 1). We created a Cas9 control construct to be used as a negative control. Plants transformed with this construct underwent transformation and regeneration with selection. As this construct lacks sgRNAs, no editing was predicted to occur at the target sites. 

Four gene-editing sgRNA constructs and a Cas9 control construct were transformed into the ‘M.26’ and ‘Royal Gala’ cultivars via *Agrobacterium*-mediated genome insertion. We obtained a total of 44 transformed events (Table 2). Here, transformed events were defined as independently isolated shoots after transformation and growth on selective medium, and were confirmed by PCR to have the transgene construct present (workflow and shoot verification in Figure 2). Of these events, 40 were in ‘M.26’ and 4 were in ‘Royal Gala’. We observed 6 total Cas9 control events and 38 total CRISPR events, with the number of events per construct and cultivar ranging from 0 to 13. 

We initially used dual-allele amplification of each target gene, meaning amplification directly followed by amplicon sequencing without subcloning or isolating alleles to determine if any events showed evidence of changes to target genes. Primers used are listed in Appendix A. Analysis of the sequence peaks and alignments with the wild-type sequence showed a variety of outcomes (Appendix A). These included events with no changes to target sites (no changes in alignment and single sequence peaks), events with identical bi-allelic changes (changes in alignment and single sequence peaks, indicating identical alleles), and events with allelic differences (changes in alignment and double sequence peaks, indicating non-identical alleles). As these findings indicated there were changes to at least some target sites in some of our events, we decided to pursue allele-specific sequencing. 

We then used dual-allele amplification of each target gene followed by subcloning of amplicons to separate alleles from each gene. Subcloned fragments were sent for DNA sequencing and the results aligned with our initial target-site sequences. Cloning of both alleles was verified by the number of amplicons present on the gel and by allele-specific single nucleotide polymorphisms (SNPs). 

We observed six Cas9 control events, which had the Cas9 gene added but no sgRNAs to direct Cas9 to target sites. For these events, analysis of the *MADS15* and *MADS221* genes showed no changes to the target genes. Of our 38 CRISPR-Cas9 events, 34 showed changes to at least one allele (for one or more target genes), while 4 events had no changes to any sites, for an overall event-targeting rate of 89% (Table 3). We found that 14 events had changes to all four alleles (changes in both alleles of *MADS15* and both alleles of *MADS221*), with a rate of 37% for all events. The remaining 20 events were a mix of WT and edited alleles. Complete information on each event and each allele can be found in Appendix A.

### 2.2. Mutation Types and Frequency in Target Genes 

We also analyzed the data in terms of editing by sgRNA. Three of the sgRNAs (sgRNA1, sgRNA2, sgRNA3) were designed to target both *MADS15* and *MADS221*; they were termed dual-gene targeting, while two of the guide RNAs were specific to either *MADS15* (sgRNA15) or *MADS221* (sgRNA22, Appendix A). Examination of targeting by guide RNA showed that target editing by sgRNA ranged from 37–68% of targets changed, with sgRNA3 giving the highest rate of editing (Table 4). 

Each of the four editing constructs contained two sgRNAs (Figure 1). Examination of editing frequency by construct showed that one construct led to edits in all events obtained, while the other two constructs gave editing rates of 75–93% (Table 5).

Examination of mutation types showed that most changes were large deletions greater than 16 base pairs (bp); these accounted for 36–82% of edits observed from each construct (Table 6). Small deletions were also common (11–21%), as were small insertions (0–19%). All constructs led to a variety of mutation types, with both large and small deletions observed.

We examined both of our target genes for changes. Alignment of sequences showed a variety of mutations at a given target site for each gene. Small mutations tended to cluster around each sgRNA, while larger deletions spanned the region flanked by two sgRNA sites (Figure 3 and Figure 4).

We used the ExPASy peptide prediction program [65] followed by peptide alignment to illustrate possible peptides produced by our edited alleles. Some mutations were in-frame and led to the alternation of a portion of the predicted peptide, with most of the sequence intact. Other mutations led to predicted frame shifts and severe alterations in predicted peptide sequences, which often led to predicted early stop codons (Figure 5).

No obvious differences were noted between CRISPR-edited lines or within any line in terms of growth or overall appearance. Apple trees require several years to overcome juvenility, during which time they have no capacity to bloom. These trees are now entering maturity, as evidenced by extremely preliminary data: a single bloom from Line T532, which shows the expected phenotype of doubled petals and altered sexual organs (Appendix A). In contrast, the M.26 parental cultivar is a single-flowered variety with well-formed stamens. 

## 3. Discussion

Targeted genome editing in fruit crops such as apple has the potential to create high-value traits or remove unwanted features. For apples and other pomme fruits, some desirable traits could include reduced invasive potential, non-browning fruit, improved tolerance to abiotic and biotic stresses, and altered tree form. This sort of editing can also be highly useful in research, allowing for the creation of specific genetic knockouts. Here, we selected the gene targets *MADS15* and *MADS221*, as they appear to be *AG* homologs in domestic apple. Prior work suppressing these genes via RNAi found that reduced AG-like gene function led to the conversion of anthers to petals and reduced seed set [63,66]. Thus, these genes may be useful targets for engineering floral sterility. 

At the time this project was initiated in 2015, the efficiency of CRISPR-Cas9 gene editing in trees was still being discovered. Hence, we created double sgRNA constructs with redundant-gene targeting to try and increase the chances of edits to target sites. Additionally, using two sgRNAs would increase the likelihood of larger deletions. We also created a Cas9 control construct that lacked any guide RNAs to serve as our negative control, as suggested in CRISPR-Cas9 genome editing protocols [67]. We observed no target-site changes in our six Cas9 control events, as would be expected of a lack of genome editing in the absence of sgRNAs. All four editing constructs successfully led to the creation of events with changes to one or more of the target gene alleles. We found that the overall editing rate was high, with most events (34, 89%) showing edits to at least one target allele, and 14 events (37%) showed editing of all four alleles. These findings are similar to initial CRISPR-Cas9 gene-editing work in apple showing efficient target-site changes, with an editing rate of 31.8% for PDS [22] and a rate greater than 85% for PDS and TLF1 [16]. Some of this variation may be due to the different cultivars used, the gene or genes targeted, or to the regeneration and transformations methodologies used. 

We found a variety of mutation types in our CRISPR-Cas9 events. Most mutations were large deletions spanning the region between the two sgRNAs, indicating activity at both sgRNA sites. This trend was observed for all four constructs tested. Some alleles had a mixture of small insertions and deletions. The small deletions and insertions we observed were similar to what was found previously for double sgRNA modification in domestic apple [16] and single sgRNA editing in domestic apple [22]. Our finding of many larger deletions spanning the sgRNA sites is different from these prior studies in apple, but is similar to work in *Populus*, where the use of two sgRNAs led to frequent larger deletions [68]. Unlike prior work using two sgRNAs in apple [16], we did not observe inversions at our target sites. Analysis of editing efficiency by construct and sgRNA showed that all four gene-editing constructs we developed were effective at achieving gene changes. As has been observed before in apple [22], sgRNAs were variable in their effectiveness, but all led to edits. 

Analysis of predicted peptides for our events showed a range of predicted outcomes. Some DNA changes had minor impacts on the predicted peptide sequence, such as one deletion followed by one substitution, or the deletion of three amino acids. Other mutations that altered the reading frame led to early stop codons. Assessment of the functional impact of these changes will require waiting until most trees mature and flower. Currently, a single experimental tree has reached maturity. This early bloomer has given a preview of flowers from an event with changes to all four target alleles. Our trees have four total alleles and a variable number of mutated alleles and mutation types, ranging from no mutations to all four alleles changed. It will be interesting to see how this spectrum of mutations influences floral phenotypes. 

We did not observe evidence of chimerism in the events characterized in this study, as we did not detect more than two alleles for each gene in any given event. Chimerism after organogenic regeneration is a concern for the study and the usage of CRISPR-Cas9-edited plants [69]. A prior study in apple found that 88% of PDS-targeted events and 100% of TFL1.1-targeted events were chimeric, as were 80% of TFL1.1-targeted events in pear [16]. Indeed, many initial studies targeting *PDS* in fruit crops observed chimeric explants, as evidenced by the presence of striped, variegated, or sectored regenerated plantlets (Table 1). However, in plants with *PDS* targeting, loss of gene function is detrimental to plant growth, with albino cells requiring nutritional support from the growth medium or from photosynthetic cells, and plantlets containing at least some green photosynthetic cells would have a growth and health advantage over their fully albino counterparts [70]. Thus, targeting *PDS* may give an overestimate of the general rate of chimerism that occurs in transformation and regeneration with gene editing. The final outcome of chimerism in a regenerated plant could arise from a genetically variable population of cells forming the final plantlet or could be due to ongoing CRISPR-Cas9 activity over time in the plantlet. An investigation of regenerated events derived from initially edited lines showed the presence of new genetic changes to target sites [69]. Thus, it appears that retained stably-integrated CRISPR editing constructs may have continued activity, or regenerated plants could arise from a subpopulation of cells from a chimeric parental plant. However, studies targeting tree genes other than *PDS* found a relatively low rate of chimerism. For example, a large-scale study of floral gene targeting in *Populus* characterized 591 events from a variety of CRISPR-Cas9 constructs and found a global chimerism rate of 1.3%, with most constructs giving 0% chimeric events [68]. Another study in *Populus* targeting seven tandemly-arrayed *Nucleoredoxin1* (*NRX1*) alleles analyzed 42 individual events and uncovered a wide range of mutation types, including complex fusions and rearrangements, but no evidence of chimerism [71]. In eucalyptus, targeting of the key floral gene LEAFY showed no evidence of chimerism in the 68 events studied [72].

Overall, we found that CRISPR targeting of our two genes simultaneously led to a variety of edited outcomes, with some events fully WT and others edited at all alleles. Most mutations were large deletions, but a wide spectrum of mutation types was observed. As events varied in the number and severity of gene changes, they represent a highly variable population for future study. As our targeted genes are key for overall floral morphology, future work will hopefully include complete assessment of floral form, fertility, and fruits. 

## 4. Materials and Methods

The draft version of the apple genome [62] was used to obtain sequences for *MADS15* (Gene ID MDP0000324166, Gene symbol LOC103434929, GenBank cDNA Accession AJ251118, Phytozome Gene Identifier MD07G1022500) and *MADS221* (Gene ID MDP0000250080, GenBank cDNA Accession HM122606, Phytozome Gene Identifier MD10G1271000). These data were used to design primers for amplifying and sequencing these genes from the ‘M.26’ and ‘Royal Gala’ cultivars that we selected for transformation and regeneration. Primers used for gene sequencing are found in Appendix A. The ‘M.26’ cultivar is commonly used as a dwarfing rootstock [73], while ‘Royal Gala’ is used for fruit production [74]. The cultivar-specific gene data were used to select guide RNA sites. Selected sites were 20 bp in length and adjacent to the sequence NRG, based on the cut-site preference for the Cas9 variant used. 

Construct assembly

The Cas9-only control construct was created previously [68]. Double sgRNA constructs were created as per previously published methods [68]. In brief, two oligos were ordered for the assembly of each sgRNA (oligos for sgRNA construction are listed in Appendix A). Oligos were phosphorylated, annealed, and ligated into the vector psK-AtU626. Final assembled Cas9 sgRNA cassettes were ligated into the plant transformation vector pK2GW7. 

Apple transformation and event validation

Clonally propagated apple (*Malus* x *domestica*) ‘M.26’ and ‘Royal Gala’ leaves underwent *Agrobacterium*-mediated transformation as described previously [75]. In brief, young leaves were harvested from axenic cultures, lightly wounded with non-traumatic forceps, inoculated with *Agrobacterium tumefaciens* strain EHA105 (pCH32) harboring the desired CRISPR constructs, and placed on co-cultivation media in the dark for three days. The leaves were then removed from the media, rinsed, cut into approximately 1 mm wide strips, and placed with the abaxial side down into regeneration media in the dark for three weeks. The regeneration plates were then placed in low light (Figure 2A,B) and regenerants (Figure 2C) transferred to shoot proliferation media (Figure 2D). The resulting regenerants were verified via PCR from tissue-culture plantlet leaves (Figure 2D) using the following primers: *Malus* x *domestica elongation factor 1* (*MdEF1*, F5′-GACATTGCCCTGTGGAAGTT-3′, R 5′-GGTCTGACCATCCTTGGAAA-3′), used to assess competence of DNA for PCR; VirG (F 5′-GCCGGGGCGAGACCATAGG-3′, R 5′-CGCACGCGCAAGGCAACC-3′), used to assess whether there is residual, endophytic *Agrobacterium* contamination; and Cas9 and nos terminator specific primers to check for transgene presence (Cas9_end_F2: 5′-CCTACAACAAGCACCGGGAT-3′ and Cas9_tnos_R2: 5′-AACGATCGGGGAAATTCGAG-3′). Explants were propagated in tissue culture at the USDA-ARS-NEA-AFRS facility (Kearneysville, WV, USA), essentially as per published methods [76,77] with root induction [78]. Own-rooted young trees were transferred to soil and grown in a greenhouse with supplemental lighting (Na vapor) to maintain the day length at 16 h, and a maximum/minimum temperature range of 35/20 °C. Trees were watered daily and fertilized regularly (Appendix A). 

Target-site cloning and analysis

Initial amplicons of *MADS221* and *MADS15* genes were isolated from extracted DNA and amplified using a Q5 polymerase (New England BioLabs^®^ Inc. M0491S, Ipswich, MA, USA) with gene-specific primers (*MADS221*: Forward 5′ ATGGCCAATGAAAACAAATCC 3′, Reverse 5′ CTGTTGTTGGCATACTCATAGAGG 3′; *MADS15*: Forward 5′ ATGGCCTATGAAAGCAAATCC 3′, Reverse 5′ CTATTGTTGGCATACTCATAGAGG 3′). PCR was performed with a SimpliAmp™ Thermal Cycler (Applied Biosystems A24812), and annealing temperature was set to 58 °C for 33 cycles. Sterile DI H_2_O was used for a negative control. The PCR target sequence was 230 bp for each gene. For subcloning, 4 µL of PCR product was set aside, and the remainder was combined with loading dye (40% sucrose, 30% glycerol, 2.0% Orange G, 10× concentration, used at 1X) for visualization on a 1% agarose gel with a 1 Kb DNA Ladder (New England BioLabs^®^ Inc. N3232L, Ipswich, MA, USA) or 1 Kb Plus DNA Ladder (New England BioLabs^®^ Inc. N3232L, Ipswich, MA, USA). Gels were visualized using a GelDoc Go Imaging System (BioRad Laboratories Inc.). Larger amplicons were obtained with the same method using primers MADS15 new F1 (5′ TTTCATTTGTTTCTGCAAGTTTC 3′) and new R1 (5′ CAAGTAAATTGAAGCAATAATATTACCTA 3′), and MADS221 new F2 (5′ GTTGGTGATCAAGAATATAGTAATTG 3′) and new R1 (5′ AGCAAGAAAATTGAAGCAATAATATTATTA 3′). Retained PCR products were subcloned into the vector Zero Blunt^®^ TOPO^®^ PCR Cloning Kit for Sequencing (Invitrogen™ 45-0031, Waltham, MA, USA) according to kit instructions with a 30 min initial incubation time. Cloning reactions were transformed into chemically competent *E. coli* cells (One Shot™ TOP10 Chemically Competent *E. coli*, Invitrogen™ C404010, Waltham, MA, USA) as per kit instructions. Transformed cells were divided among two LB + kanamycin (KAN) plates (1.0% tryptone, 0.5% yeast extract, 1.0% NaCl, 1.5% agar powder, 0.05% kanamycin) with 50 µL on one plate and the remainder (~300 µL) on the second plate. Plates were allowed to dry on the benchtop until all liquid was absorbed and were then incubated overnight at 37 °C (VWR™ 1515E, Radnor, PA, USA). Afterwards, the plates were wrapped in parafilm and stored at 4 °C for colony analysis. 

Colony analysis

Eight colonies were selected from the transformed plates and resuspended in a 1.5 mL centrifuge tube with 100 µL of LB+KAN (1:1000). The colony broth was pipetted or vortexed (Fisher Analog Vortex 02215414, Waltham, MA, USA) and then shaken for 1 h at 37 °C. Colony PCR was performed using the colony broth with EconoTaq^®^ DNA Polymerase (Lucigen 30031-1, Middleton, WI, USA) and the corresponding gene primers (*MADS221* or *MADS15*) to allow for the amplification of cloned fragments. An annealing temperature of 56 °C was used for 33 cycles. Purified TOPO cloning vector with a cloned *MADS15* fragment was used as a positive control, and sterile DI H_2_O was used as a negative control. Loading dye was added to the PCR product and visualized using a 1% agarose gel and 1 Kb DNA Ladder. Generally, 4 colonies with amplified fragments were selected from each sample for each gene for overnight liquid culture and plasmid extraction. Sample selection was based on relative band size of the product(s), and if colonies had different sized bands, then colonies with different sized products were selected for analysis. Next, 30 µL of the colony broth was used to inoculate a 5 mL culture of LB+KAN (1:1000), which was shaken at 37 °C overnight.

Plasmid Extraction and Digestion

Cells from the liquid culture were then pelleted using a Sorvall Legend Micro 21 Centrifuge (ThermoFisher Scientific, Waltham, MA, USA), and the DNA was extracted using a Monarch^®^ Plasmid Miniprep Kit (New England BioLabs^®^ Inc. T1010L/S, Ipswich, MA, USA) following kit instructions. DNA was quantified using a NanoDrop 2000 (ThermoFisher Scientific E112352, Waltham, MA, USA). Extracted DNA was then used for an enzyme digest with EcoRI (New England BioLabs^®^ Inc. R0101S/L, Ipswich, MA, USA) to confirm successful gene fragment cloning, as the TOPO vector has EcoRI sites flanking the MCS, and no EcoRI sites are located in the target gene fragments. Solutions were incubated for at least 2 h at 37 °C and then visualized using a 1% agarose gel. Based on visualization results of the enzyme digest, plasmids were selected for sequencing. 

DNA Sequencing and Peptide Prediction

The selected plasmid samples were mixed to a final DNA quantity of 400 ng in 10 µL with sterile DI H_2_O and sent out for Sanger Sequencing with GENEWIZ (Azenta Life Sciences, Burlington, WI, USA) using the M13F primer provided by Azenta. Sample sequences were compared with the WT gene form from the proper cultivar using the alignment tools available through the blastn suite [79]. Mutations were catalogued digitally based on size and type (i.e., 1 bp insertion, 15 bp deletion, etc.). Peptide predictions were made using the ExPASy Translate tool [65]. Alignments of predicted peptides were created using Clustal Omega [80]. 

## 5. Conclusions

CRISPR-Cas9 gene editing can be successfully applied to obtain dual targeting in domestic apple. The overall efficiency varies both by sgRNA and by gene target, with some trees retaining WT alleles. As the CRISPR-Cas9 system was stably transformed and retained in the trees, a long-term goal is surveying for continued Cas9 activity and allele targeting, along with analysis of vegetative performance and floral form.

## Figures and Tables

**Figure 1 plants-12-03693-f001:**
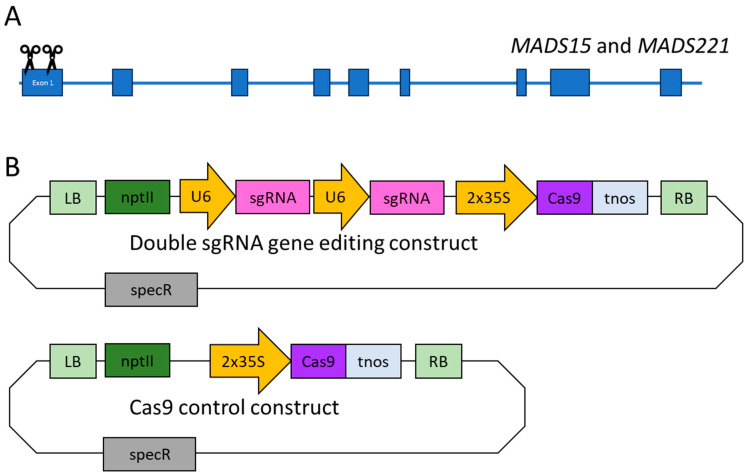
Location of sgRNAs in *MADS15* and *MADS221* and construct diagrams. (**A**) Schematic of targeting locations in *MADS15* and *MADS221*. Exons are shown with boxes, intron with lines, and scissors represent general locations of sgRNAs. The *MADS15* and *MADS221* genes are predicted to have the same overall gene structure. As targeting constructs had two sgRNAs, two scissors are shown to indicate the approximate cut sites in exon 1. The total length of *MADS15* and *MADS221* is ~3102 base pairs. Gene structure is from the alignment of coding and genomic sequences obtained from Phytozome. (**B**) Construct diagrams. We created four double sgRNA constructs for genome editing (the general structure of which is shown here) and a Cas9-only control construct. LB, T-DNA left border; nptII, neomycin phosphotransferase II gene for kanamycin selection in plants; U6, U6-26 gene promoter from *Arabidopsis thaliana*; sgRNA, locations of sgRNAs; 2 × 35S, double 35S gene promoter from *Cauliflower mosaic virus*; Cas9 human, codon-optimized Cas9 from *Streptococcus pyogenes*; tnos, nopaline synthetase terminator from *Agrobacterium tumefaciens*; RB, T-DNA right border; specR, spectinomycin resistance gene for selection in bacteria. Promoter directionality is shown with arrows.

**Figure 2 plants-12-03693-f002:**
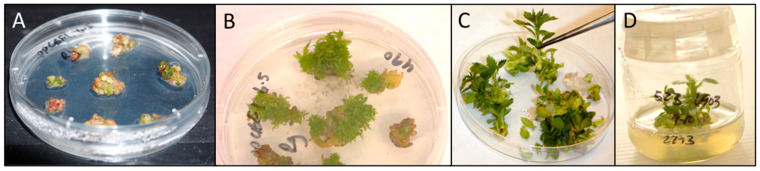
Images from transformation and regeneration. (**A**) Callus forming on regeneration plates after co-cultivation; (**B**) shoots forming after 4 weeks in the light on regeneration medium; (**C**) larger shoots were chosen and transferred to shoot proliferation medium; (**D**) regenerants in shoot proliferation medium; leaves from this stage were used for transformation event confirmation via PCR.

**Figure 3 plants-12-03693-f003:**
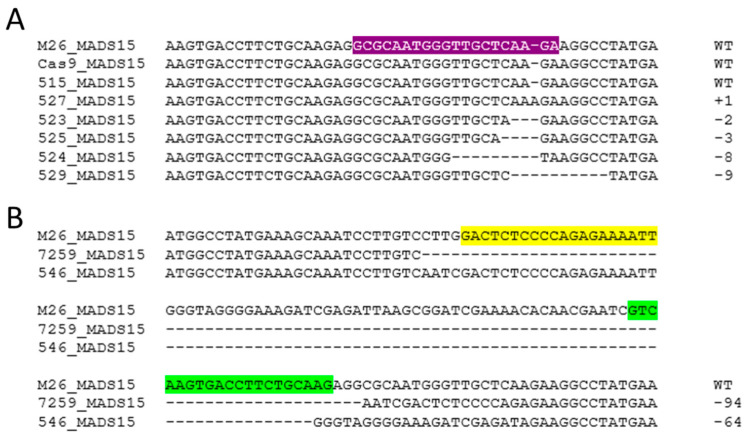
Examples of edits in *MADS15.* (**A**) Examples of small edits in *MADS15* from different independent events at the universal sgRNA3 site shown in purple. (**B**) Examples of large deletion events obtained with sgRNA1 shown in green and those obtained with sgRNA2 shown in yellow. Numbers indicate the amount of inserted (+) or deleted (−) bases.

**Figure 4 plants-12-03693-f004:**
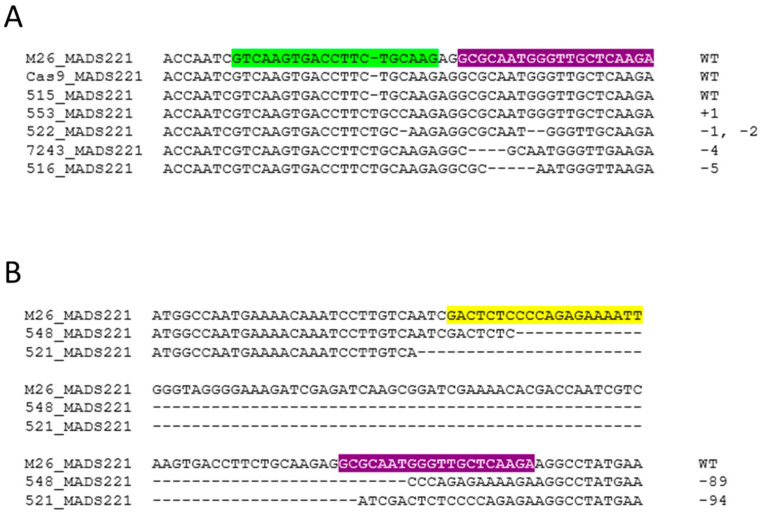
Examples of edits in *MADS221.* (**A**) Examples of small edits in *MADS221* from different independent events, with the sgRNA1 site shown in green and the sgRNA3 site shown in purple. (**B**) Examples of large deletion in events obtained at the sgRNA2 site shown in yellow and at the sgRNA3 site shown in purple. Numbers indicate the amount of inserted (+) or deleted (−) bases.

**Figure 5 plants-12-03693-f005:**
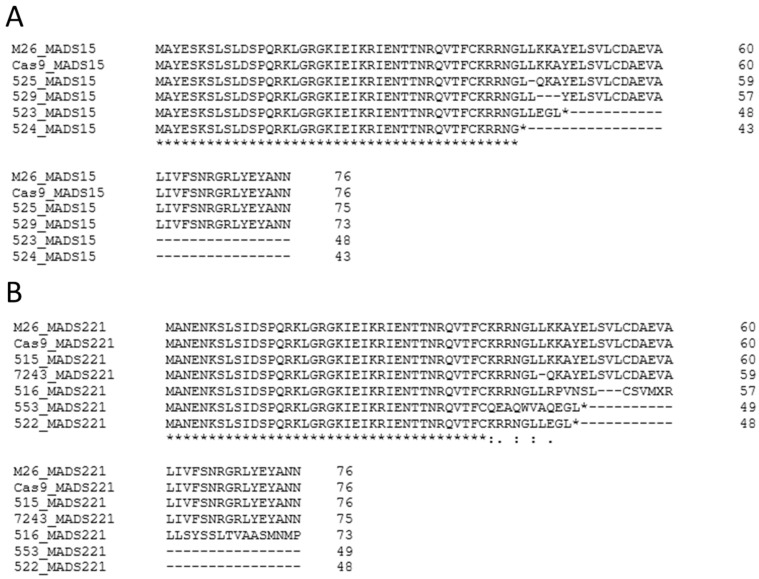
Predicted peptides of *MADS15* and *MADS221* from alleles with small indels. (**A**) Alignment of predicted peptides for the beginning of *MADS15* for WT, Cas9, and four different events with small indels. (**B**) Alignment of predicted peptides for the beginning of *MADS221*. Dashes indicate absent amino acids, asterisks in alignments indicate stop codons, asterisks under alignments indicate exact sequence matches, colons and periods under alignment indicate partial matches, and numbers on the far right indicate the length of the predicted fragment.

**Table 2 plants-12-03693-t002:** Transformation events obtained per construct and cultivar. We obtained a total of 44 transgenic events, 6 Cas9 control events and 38 CRISPR events, from our four different CRISPR constructs. Event refers to an independent plant transformation occurrence.

Construct	Cultivar	Transformed Events
Cas9	‘M.26’	5
Cas9	‘Royal Gala’	1
Total Cas9 events	Both	6
sgRNA1 sgRNA2	‘M.26’	8
sgRNA 1 sgRNA2	‘Royal Gala’	1
sgRNA15 sgRNA3	‘M.26’	6
sgRNA15 sgRNA3	‘Royal Gala’	1
sgRNA3 sgRNA2	‘M.26’	8
sgRNA3 sgRNA2	‘Royal Gala’	0
sgRNA22 sgRNA3	‘M.26’	13
sgRNA22 sgRNA3	‘Royal Gala’	1
Total CRISPR events ‘M.26’	‘M.26’	35
Total CRISPR events ‘Royal Gala’	‘Royal Gala’	3
Total CRISPR events	Both	38
Total transgenic events (Cas9 and CRISPR)	Both	44

**Table 3 plants-12-03693-t003:** Summary of whether editing occurred in each target gene allele and frequencies. We analyzed all 38 CRISPR-Cas9 events for changes in alleles of each target gene. Events with no changes in a given target gene and alleles were designated WT for that gene, events with one WT and one altered allele were designated heterozygous (het) for that gene, and events with alterations to both alleles of a gene were designated dual-edited for that gene. Dual-edited indicates either two different edited alleles, or identical edited alleles.

Outcome Per Gene		Events (Number and %)
*MADS15*	*MADS221*	
WT	WT	4 (11%)
WT	het	2 (5%)
het	WT	3 (8%)
WT	Dual-edited	3 (8%)
Dual-edited	WT	0 (0%)
het	het	4 (11%)
Het	Dual-edited	5 (13%)
Dual-edited	het	3 (8%)
Dual-edited	Dual-edited	14 (37%)
Fully WT		4 (11%)
Mix of WT and edited		20 (53%)
Fully edited		14 (37%)

**Table 4 plants-12-03693-t004:** Mutation frequency by sgRNA and *MADS*-like genes. We used three sgRNAs that could target both genes in each cultivar. These dual-gene targeting sgRNAs (sgRNA1, sgRNA2, sgRNA3) could target both alleles of *MADS15* and *MADS221*. The *MADS15*-specific (sgRNA15) and *MADS221*-specific (sgRNA22) sgRNAs could each target either *MADS15* or *MADS221* in both cultivars. As guide RNAs were used in pairs to create each construct and variable events were obtained for each construct, the number of potential targeting sites per guide RNA varies. In total, our sgRNAs had 262 possible sites. Targeting was defined as at least one change from the cultivar-specific WT sequence for a given allele.

sgRNA	Total Sites	Targeting (Number and %)
Dual-gene targeting		
sgRNA1	36	17 (47%)
sgRNA2	68	26 (38%)
sgRNA3	116	79 (68%)
*MADS15* specific		
sgRNA15	14	10 (71%)
*MADS221* specific		
sgRNA22	28	17 (61%)
All guide RNAs		
sgRNA1, sgRNA2, sgRNA3, sgRNA15, sgRNA22	262	149 (57%)

**Table 5 plants-12-03693-t005:** Editing by CRISPR construct. We created one Cas9-only control and four different gene-editing CRISPR-Cas9 constructs, with 7 to 14 events per construct, combined between cultivars. Editing refers to any change from the WT sequence.

Construct	Number of Transformation Events	Number of Edited Events	% Editing
Cas9 control	6	0	0
sgRNA1 sgRNA2	9	8	89
sgRNA2 sgRNA3	8	6	75
sgRNA15 sgRNA3	7	7	100
sgRNA22 sgRNA3	14	13	93

**Table 6 plants-12-03693-t006:** Frequencies of deletions, insertions, or small indels. Insertion and deletion mutations were classified as small if they involved 15 or fewer bp, and large if they were 16 or more bp. The percentages are calculated as the frequency of that mutation type for all changed alleles for a construct. The most common mutation type per construct is bolded.

Construct	Total Events (N)	Total Alleles (N)	Total Edited Alleles (N) %	Large Deletion (N) %	Small Deletion (N) %	Small Insertion (N) %	Mix of Small Indels (N) %
sgRNA1 sgRNA2	9	36	17 (47%)	**14 (82%)**	3 (18%)	0 (0%)	0 (0%)
sgRNA2 sgRNA3	8	32	16 (50%)	**10 (63%)**	3 (19%)	3 (19%)	0 (0%)
sgRNA15 sgRNA3	7	28	22 (79%)	**14 (50%)**	3 (11%)	5 (18%)	0 (0%)
sgRNA22 sgRNA3	14	56	43 (77%)	**20 (36%)**	12 (21%)	4 (7%)	7 (13%)

## Data Availability

The data presented in this study are available in the article or Appendix A.

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
