# Peer review of "Apple CRISPR-Cas9—A Recipe for Successful Targeting of *AGAMOUS*-like Genes in Domestic Apple"

_plants, 2023, doi:10.3390/plants12213693_

Round 1
Reviewer 1 Report
Comments and Suggestions for Authors
The manuscript describes gene editing of the agamous gene in apple with a focus on dual editing using gRNAs that are homologous to different AG genes in apple. Although gene editing is not new in apple there is novelty in the dual editing approach.
Suggested improvements include:
The abstract could be improved as a standalone description. The differences between Cas9 and independent CRISPR-cas9 events (ln 20-21) is also not immediately clear.
In the introduction it is suggested (ln 40) that both conventional and GE approaches can sometimes lead to the same desired outcomes. This is not supported by the references. For instance, although conventionally bred purple tomato is available the anthocyanin is restricted to the skin. In the GE one cited the anthocyanin is also high in the flesh leading to a potentially more desirable (health and storage) outcomes.
The contents of Table 1 are somewhat repeated in the text and not complete. Several examples of editing PDS1 in apple are given but more interesting recent "applied work" is not mentioned i.e., MdPGT1, MdSPL6 etc. Published applied work in other fruit crops such as kiwi are also absent.
The introduction could be a little more concise, there is a sizable component relating to a review of fruit tree improvement in general.
In the results section (ln 159) it is not clear what the "four SgRNA constructs" were - they don't appear to be mentioned earlier?
Ln 160 - how are transformed events defined? Was this based on antibiotic selection or were molecular studies to confirm the presence of the editing associated genes carried out?
Ln 170 what is "dual allele amplification"? Is this amplification and sequencing of a single band?
Ln 291 what peptide prediction program?
The first part of the discussion rephrases the introduction.
Ln 354 makes interesting comments about the lack of chimerism observed in the study and discusses this. How was this data derived? There seems to be no information on this in the manuscript.
Were any off-targeting effects detected with the MADS221-1 and MADS15-1 specific guides?
The early part of the materials and methods seems appropriate given that the methodology has been published before. The sections on Target site cloning and analysis/Plasmid extraction and digestion in contrast are perhaps overly long for standard methodology.
Ln468, the use of only 4 colonies to characterise the nature of editing frequency seems quite low. If chimeric editing were present there seems to be a good chance to miss this with so few sequences.
Were the guide RNA sequences selected using a prediction programme or by eye?
Comments on the Quality of English Language
The English is of a good standard.
Author Response
The manuscript describes gene editing of the agamous gene in apple with a focus on dual editing using gRNAs that are homologous to different AG genes in apple. Although gene editing is not new in apple there is novelty in the dual editing approach.
Suggested improvements include:
The abstract could be improved as a standalone description. The differences between Cas9 and independent CRISPR-cas9 events (ln 20-21) is also not immediately clear.
We have added the word “control” after the Cas9 events and hope this change here and additional changes in the main text will help to clarify the distinction between Cas9 and CRISPR-Cas9 events.
In the introduction it is suggested (ln 40) that both conventional and GE approaches can sometimes lead to the same desired outcomes. This is not supported by the references. For instance, although conventionally bred purple tomato is available the anthocyanin is restricted to the skin. In the GE one cited the anthocyanin is also high in the flesh leading to a potentially more desirable (health and storage) outcomes.
The purpose here was to show that the same trait (purple fruit) could be obtained through both approaches.
The contents of Table 1 are somewhat repeated in the text and not complete. Several examples of editing PDS1 in apple are given but more interesting recent "applied work" is not mentioned i.e., MdPGT1, MdSPL6 etc. Published applied work in other fruit crops such as kiwi are also absent.
We thank the reviewer for the suggestions to include the Miranda et al 2023 work on apple PGT1, and Li et al 2023 on MdSPL6. Both of these examples have been added to Table 1. Work in kiwi is cited in Table 1. The text was meant to summarize some of the key findings, not list everything from the table.
The introduction could be a little more concise, there is a sizable component relating to a review of fruit tree improvement in general.
This section was included as it was a good fit for the Special Issue of the journal. It does not cover many aspects of fruit tree improvement in general, such as marker assisted breeding, random mutagenesis, and more.
In the results section (ln 159) it is not clear what the "four SgRNA constructs" were - they don't appear to be mentioned earlier?
This section of the text beginning line 147 now reads “We designed a set of single guide RNAs (sgRNA) to target our two genes simultaneously (Supplementary Figure 1). The sgRNA sites were located in the first exon to hope-fully disrupt protein coding (Figure 1). Constructs had two sgRNAs each to increase tar-get gene editing frequency and potentially create larger deletions (Figure 1). We created 4 different sgRNA constructs, each with a different pairing of sgRNAs (pairings and sgRNA sequences are in Supplementary Figure 1). We used two sgRNAs in each gene editing construct to increase target gene editing frequency and potentially create larger deletions (Figure 1).”
Ln 160 - how are transformed events defined? Was this based on antibiotic selection or were molecular studies to confirm the presence of the editing associated genes carried out?
We have added the wording starting line 172, “Here, transformed events were defined as independently isolated shoots after transformation and growth on selective medium, and were confirmed by PCR to have the transgene construct present.” Precise details on the primers used can be found in the methods section.
Ln 170 what is "dual allele amplification"? Is this amplification and sequencing of a single band?
The wording here beginning line 193 was changed to “We initially used dual allele amplification of each target gene, meaning amplification directly followed by amplicon sequencing without subcloning or isolating alleles to determine if any events showed evidence of changes to target genes.”
Ln 291 what peptide prediction program?
This line has been changed to “We used the ExPASy a peptide prediction program [63] followed by peptide alignment to illustrate possible peptides produced by our edited alleles.” This is now Line 319.
The first part of the discussion rephrases the introduction.
Yes, we felt a quick summary to frame the work was suitable before starting on discussion of our findings.
Ln 354 makes interesting comments about the lack of chimerism observed in the study and discusses this. How was this data derived? There seems to be no information on this in the manuscript.
This section has been clarified as “We did not observe evidence of chimerism in the events characterized in this study as we did not detect more than two alleles for each gene in any given event.”. This is now Line 398.
Were any off-targeting effects detected with the MADS221-1 and MADS15-1 specific guides?
We did not examine off-target sites as part of this study.
The early part of the materials and methods seems appropriate given that the methodology has been published before. The sections on Target site cloning and analysis/Plasmid extraction and digestion in contrast are perhaps overly long for standard methodology.
We agree that this part of the methods is a bit verbose. It was written by undergraduate students then revised for correctness as needed. It was a good exercise for them. If the manuscript needs to be shortened, we will trim this portion.
Ln468, the use of only 4 colonies to characterise the nature of editing frequency seems quite low. If chimeric editing were present there seems to be a good chance to miss this with so few sequences.
We agree that this is indeed a risk. Use of the same construct backbone for editing in poplars and eucalypts gave a very low rate of detected chimerism (1.5% of events had 3 altered alleles, 0.6 of events had 2 altered and 1 WT allele) Elorriaga et. al 2018), and no chimerism detected at all in eucalypts (Elorriaga et al. 2021). We also did not observe more than 2 bands during DNA amplification. Which, while it does not rule out the possibility of 2 similar-sized but different DNAs, shows no support for 3 very different sized alleles in any given event.
Were the guide RNA sequences selected using a prediction programme or by eye?
We selected the guide RNA sequences as per previously published methods (Elorriaga et al. 2018) and used the program ZiFit.
Reviewer 2 Report
Comments and Suggestions for Authors
Author Response
Comments and questions to the Authors
- In 2017, an updated version of the reference genome assembly of Malus Domestica Golden Delicious was published, but the authors used an older version that is incomplete and was suppressed. Is there a particular reason for that?
Initial construct work began in 2015. Subsequent comparisons of the relevant genes revealed no substantial differences. In 2017 the constructs were already created and in use. In addition, the MADS15 and MADS221 genes were also listed with their Phytozome identifiers. These identifiers are clearly from the later Malus x domestica 'Golden Delicious' genome release.
- Since multiplex sgRNAs were designed to simultaneously target genes, the potential off-target effects in transgenic mutants should be detected and analyzed carefully.
We did not examine off-target sites in our events. We acknowledge that this can be an important consideration in genome editing, but it was outside the scope of the current project.
- It is unclear from the paper why the authors used a control construct containing only Cas9, without sgRNA.
We thank the reviewer for pointing out this important part of the work. In short, this is a control construct, one form which we expect no target site changes. We have added the following text to clarify in the manuscript.
Beginning line 152, “We created a Cas9 control construct to be used as a negative control. Plants transformed with this construct underwent transformation and regeneration with selection. As this construct lacks sgRNAs, no editing is predicted to occur at target sites.
Beginning line 209, “We had 6 Cas9 control events, events which had the Cas9 gene added but no sgRNAs to direct Cas9 to target sites. “
Beginning line 366, “We also created a Cas9 control construct, which lacked any guide RNAs to serve as our negative control. We observed no target site changes in our 6 Cas9 control events, as would be expected. “
- What is the purpose of the experiments described in the first paragraphs in the Results section?
The purpose of the initial experiments were to determine the transformation efficiency (as this can vary by species and cultivar). The results section has now been sub-divided into 2a. Analysis of Transformation Efficacy and 2b. Mutation types and frequency in target genes.
- Is it necessary to use gRNA pairs to target a gene? This could increase the probability of off-target editing.
As we noted, at the time this project was started (2015) CRISPR-Cas9 genome editing was a very new area. Given the very low frequency of other genome editing techniques in trees (ZFN, TALEN) it seemed prudent to try for constructs that were likely to give at least some edits and hopefully obtain some large deletions.
Recommendations
- To divide the Results section into two subsections, for instance: Analysis of transformation efficacy; (2) Mutations types and frequency in target genes.
We thank the reviewer for this suggestion. The results are now subdivided into 2a. Analysis of Transformation Efficacy and 2b. Mutation types and frequency in target genes
- To make table headings clear enough and provide information so that the meaning of the data is clear without referring to the text.
Table headings have been revised to allow them to be better understood as stand-alone information.
Reviewer 3 Report
Comments and Suggestions for Authors
1 The methods were too simple to understand the results. For example, four sgRNA constructs were constructed, but the authors did not describe the details. The strains of Agrobacterium tumefaciens was not mentioned.
2 The regeneration process of the genome editing need to be added in the main text, not in Supplementary Figures.
3 PCR amplification and electrophoresis analysis need to be conducted to verify the gene editing events.
4 And the phenotype of the plants with gene editing was not mentioned.
Comments on the Quality of English LanguageNo comments.
Author Response
1 The methods were too simple to understand the results. For example, four sgRNA constructs were constructed, but the authors did not describe the details. The strains of Agrobacterium tumefaciens was not mentioned.
The constructs created are shown as general diagrams in Figure 1, exact sgRNA sequences are in Supplemental materials, and the details are per standard methods that have been published before. The strain of Agrobacterium was EHA105 (pCH32); this information has been added to the methods section.
2 The regeneration process of the genome editing need to be added in the main text, not in Supplementary Figures.
We agree that this is a key part of the process and thank the reviewer for this suggestion. Accordingly, we have moved our prior supplemental figure of the workflow to the main text (Figure 2).
3 PCR amplification and electrophoresis analysis need to be conducted to verify the gene editing events.
All events were analyzed by PCR amplification, sub-cloning of alleles, and allele sequencing. The sequence files for all alleles can be found in supplementary file S1.
4 And the phenotype of the plants with gene editing was not mentioned.
We have added the following text, beginning Line 324, along with a figure (Figure 5) of the single example of flowers observed to date. “No obvious differences were noted between CRISPR-edited lines or within a line in terms of growth or overall appearance. Apple trees require several years to overcome juvenility in which they have no capacity to bloom. These trees are now entering maturity, as evidenced by a single bloom from Line T532, which shows the expected phenotype of doubled petals and altered sexual organs (Figure 6).”
Reviewer 4 Report
Comments and Suggestions for Authors
This manuscript, authored by Jacobson et al., presents the application of the CRISPR/Cas9 system for targeting two AGAMOUS-like genes in apple. The authors designed gene constructs containing single guide RNAs (sgRNAs) to target either of the genes individually or simultaneously. While the research is intriguing, several areas require significant improvement in writing, data presentation, interpretation, and figures/tables. Below, I have provided comments to enhance clarity and readability:
1. The title of this paper needs to be revised for accuracy - AGAMOUS genes to AGAMOUS-like genes.
2. Carefully review the entire manuscript for the correct usage of terms such as "dual allele" and "dual gene.".
3. Line 16- can help guide use of this technology.- Specify which technology is being referred to.
4. The manuscript formatting and language require substantial improvement to enhance readability and convey the intended meaning clearly.
5. Line 137: Include information about the number of copies of MADS15 and MADS221 in the main text.
6. Line 142- Correct the term - guide RNAs (sgRNA) to single guide RNAs (sgRNAs).
7. Provide a list of guide sequences in Figure 1 or a separate table to aid in understanding the content and results.
8. Figure 1- Include the gene sizes in panel A for clarity.
9. Figure 1- Specify the distance between the two designed sgRNA binding sites or their potential cutting sites. You can also combine this information in response to comment 6.
10. Figure 1- Revise the nomenclature for promoters, terminators, genes, and other cloning modules following standard scientific norms (e.g., pU6, p2x35S, sgRNA1, sgRNA2, etc.).
11. Revise all the figures and table legends for grammar and punctuation.
12. Line 159: Describe the four sgRNA constructs as a figure or in the main text.
13. What plant/tissue stage was used for event confirmation?
14. Supple. Figure 2- Mark and highlight the information about the spacer ( 20 bp guide) and potential editing site in the figure. Use Synthego-like programs and decompose the Sanger seq data for further analysis and discuss the outcome.
15. Line 184-190 - Provide explanations of these points in the results. What do the authors mean by saying Cas9 control events? How was this confirmed? Provide the data about it. What was the purpose, and what was the outcome?
16. Clarify how the event type (mono, bi, chimera, etc.) is defined in the study.
17. Table 3- Revise the table for proper alignment.
18. Line 211-215: Clarify if this is a separate experiment or a reanalysis of the same data. Include the answer in the main text.
19. Table 3- Revise the table for proper alignment.
20. Line 254-258- Provide these Sanger seq data that is missing. Also, use Synthego-like programs and decompose the Sanger seq data for further analysis and discuss the outcome.
21. Line 265-268 - Provide the predicted amino acid changes in these Figures. Possibly trim the non-edited DNA bases and organize them with protein seq.
22. What is phenotype? Provide regenerated shoots/plantlets data in the main text and description.
Comments on the Quality of English LanguageExtensive editing of English language required
Author Response
This manuscript, authored by Jacobson et al., presents the application of the CRISPR/Cas9 system for targeting two AGAMOUS-like genes in apple. The authors designed gene constructs containing single guide RNAs (sgRNAs) to target either of the genes individually or simultaneously. While the research is intriguing, several areas require significant improvement in writing, data presentation, interpretation, and figures/tables. Below, I have provided comments to enhance clarity and readability:
- The title of this paper needs to be revised for accuracy - AGAMOUS genes to AGAMOUS-like genes.
Agreed, the title has been revised accordingly.
- Carefully review the entire manuscript for the correct usage of terms such as "dual allele" and "dual gene.".
The manuscript has been checked for the usage of these terms and corrected where necessary.
- Line 16- can help guide use of this technology.- Specify which technology is being referred to.
The wording has been changed to “the CRISPR gene-editing technology.”
- The manuscript formatting and language require substantial improvement to enhance readability and convey the intended meaning clearly.
We have made extensive edits to the text in general, and hope that these changes will help to improve the readability and understanding of the work presented.
- Line 137: Include information about the number of copies of MADS15 and MADS221 in the main text.
We have added “with two alleles per gene” to this text in Line 142.
- Line 142- Correct the term - guide RNAs (sgRNA) to single guide RNAs (sgRNAs).
This term has been corrected.
- Provide a list of guide sequences in Figure 1 or a separate table to aid in understanding the content and results.
Guide sequences are found in Supplementary Figure 1, along with their locations in the target genes.
- Figure 1- Include the gene sizes in panel A for clarity.
Gene sizes have been added to the legend for panel A.
- Figure 1- Specify the distance between the two designed sgRNA binding sites or their potential cutting sites. You can also combine this information in response to comment 6.
This diagram is a schematic, not meant to be an exact indicator of the distances between sgRNAs. Exact sites are shown in Supplementary Figure 1.
- Figure 1- Revise the nomenclature for promoters, terminators, genes, and other cloning modules following standard scientific norms (e.g., pU6, p2x35S, sgRNA1, sgRNA2, etc.).
The figure legend was revised. The 2x35S abbreviation was used before in a prior publication. The sgRNAs themselves are not numbered in the figure as we used the specific names sgRNA1, sgRNA2 for specific sgRNAs. The U6 promoter has a short abbreviation for space in the figure.
- Revise all the figures and table legends for rgrammar and punctuation.
Legends have been checked for grammar and punctuation.
- Line 159: Describe the four sgRNA constructs as a figure or in the main text.
A general schematic of a sgRNA construct is shown in Figure 1, the sequence of each sgRNA is found in supplemental figure 1.
- What plant/tissue stage was used for event confirmation?
Tissue culture leaves were used as material for event confirmation. See Figure 2 panel D (previously part of the supplementary materials).
- Supple. Figure 2- Mark and highlight the information about the spacer (20 bp guide) and potential editing site in the figure. Use Synthego-like programs and decompose the Sanger seq data for further analysis and discuss the outcome.
Supplemental Figure 2 has been revised to show the same region of the sequenced MADS15 gene from two different events from the same construct, one of which has no changes in the target site and the other which has changes in the target site. The location of the sgRNA3 match is shown for each sequence. As mentioned before, the Synthego-like program was not utilized as alleles were sub-cloned and sequenced individually.
This initial bulk sequencing was performed for a quick first look to determine if there was any evidence of changes to target sites before investing in the time and expense of sub-cloning and sequencing individual alleles. It was not meant to be a method for obtaining allele-specific data.
- Line 184-190 - Provide explanations of these points in the results. What do the authors mean by saying Cas9 control events? How was this confirmed? Provide the data about it. What was the purpose, and what was the outcome?
We thank the reviewer for pointing out this important part of the work. In short, this is a control construct, one from which we expect no target site changes. We have added the following text to clarify in the manuscript.
Beginning line 152, “We created a Cas9 control construct to be used as a negative control. Plants transformed with this construct underwent transformation and regeneration with selection. As this construct lacks sgRNAs, no editing is predicted to occur at target sites.
Beginning line 209, “We had 6 Cas9 control events, events which had the Cas9 gene added but no sgRNAs to direct Cas9 to target sites. “
Beginning line 366, “We also created a Cas9 control construct, which lacked any guide RNAs to serve as our negative control. We observed no target site changes in our 6 Cas9 control events, as would be expected. “
- Clarify how the event type (mono, bi, chimera, etc.) is defined in the study.
We did not use term “mono” to describe our events. Instead, we characterized events as WT (no changes to target alleles as compared to WT), heterozygous (one changed and one WT allele) or dual-edits (changes to both alleles). For simplicity, we reported dual-edits without parsing if the changes were identical or different. The exact information on each allele and event can be found in a supplemental file. We have added the following text to clarify on chimerism (beginning line 395) “We did not observe evidence of chimerism in the events characterized in this study as we did not detect more than two alleles for each gene in any given event.
- Table 3- Revise the table for proper alignment.
Table 3 has been revised to align better with the text, text has been centered (except for the summary categories on the bottom left). We hope these changes are what was being suggested.
- Line 211-215: Clarify if this is a separate experiment or a reanalysis of the same data. Include the answer in the main text.
This was another analysis of the data. The sentence, beginning Line 239, now reads, "We also analyzed the data in terms of editing by guide RNA."
- Table 3- Revise the table for proper alignment.
See response to identical comment 17 above.
- Line 254-258- Provide these Sanger seq data that is missing. Also, use Synthego-like programs and decompose the Sanger seq data for further analysis and discuss the outcome.
All allele sequence data is found in File S1, raw sequence reads. These data are from the sub-cloned and isolated alleles so only single peaks are present.
- Line 265-268 - Provide the predicted amino acid changes in these Figures. Possibly trim the non-edited DNA bases and organize them with protein seq.
The purpose of this figure was to show some of the variety of small and large deletions found in MADS15, the predicted peptides are displayed in a different figure as these are different data.
- What is phenotype? Provide regenerated shoots/plantlets data in the main text and description.
We thank the reviewer for this suggestion. We have added the following text, beginning Line 324, along with a new figure to address this concern. “No obvious differences were noted between CRISPR-edited lines or within a line in terms of growth or overall appearance. Apple trees require several years to overcome juvenility in which they have no capacity to bloom. These trees are now entering maturity, as evidenced by a single bloom from Line T532, which shows the expected phenotype of doubled petals and altered sexual organs (Figure 6,).”
It should be noted that all trees except for the one example of a single tree in line T532, are currently showing juvenility.
Comments on the Quality of English Language
Extensive editing of English language required
We regret that this reviewer offered no examples of where the editing or revisions are needed. All authors are native speakers of American English. The corresponding authors have senior-authored or co-authored dozens of manuscripts. While no manuscript is free of errors, it is notable that none of the other three reviewers checked the box indicating that extensive English revisions were required. We have made many small changes to the text to clarify on naming of alleles, genes, guide RNAs and so forth.
Round 2
Reviewer 3 Report
Comments and Suggestions for Authors
1 Figure 1A, the sequence of MADS15 and MADS221 is same? What is the difference between MADS15 and MADS221 from different cultivars? Figure 1B the authors created 4 different sgRNA constructs, but there was 1 in Figure 1B. Moreover, most of the results 2a was about methods not results.
2 In Figure 6 there was no control.
Overall, the results parts were still logic confusion, and the authors did not answer the reviewer’ questions.
Comments on the Quality of English LanguageNo comments.
Author Response
We thank the reviewer for reading and providing feedback on the changed manuscript. Below are our responses to each comment. Changes can be found in the changes-tracked manuscript as well.
1 Figure 1A, the sequence of MADS15 and MADS221 is same?
The structure of the two genes is predicted to be conserved.
What is the difference between MADS15 and MADS221 from different cultivars?
The differences are found in the sequences as shown in supplementary figure 1. There are a few single nucleotide differences.
Figure 1B the authors created 4 different sgRNA constructs, but there was 1 in Figure 1B.
Yes, there were indeed 4 different sgRNA constructs, the exact combinations of sgRNAs are found in the construct names and are listed in the supplementary figures. The image shown in 1B is a general construct diagram. As all the sgRNA constructs have the same backbone (with sgRNAs being the only difference) showing all 4 would be quite repetitive. For clarity, we have changed this part of the figure legend to read “We created four double sgRNA constructs for genome editing (the general structure of which is shown here) and a Cas9 only control construct.” (lines 162 and 163).
Moreover, most of the results 2a was about methods not results.
Yes, this figure indeed shows the methodology. It was moved from supplementary to main text at the suggestion of another reviewer.
2 In Figure 6 there was no control.
At this point in time only one tree has reached maturity. We present these findings as an interesting preview of the floral phenotype, not a definitive study. We have added the following text to the discussion (lines 395-397) “Currently, a single tree has reached maturity. This early bloomer has given a preview of flowers from an event with changes to all four target alleles.”
Overall, the results parts were still logic confusion, and the authors did not answer the reviewer’ questions.
We are unsure of which questions are unanswered and put in a good faith effort to edit the manuscript according to prior comments of this reviewer and those of the other three reviewers. We hope that the current version is improved and meets reviewer approval.
Round 3
Reviewer 3 Report
Comments and Suggestions for Authors
1 Figure 1A,MADS15 and MADS221,MADS15 or MADS221?
2 The authors obtained a total of 44 transformed events which were confirmed by PCR, how to define the “transformed events”, how to do it, there was no results about the PCR.
3 The results on “transformed events” and “Mutation types and frequency in target genes” were confusing. Major revision is needed for the results parts.
4 Figure 6, if there was no control, the results were not credible.
Author Response
We thank the reviewer for taking the time to review this revised manuscript again. Below is our point-by-point response. We have provided a changes-tracked and clean copy of the revised manuscript.
1 Figure 1A,MADS15 and MADS221,MADS15 or MADS221?
We used the “and” to indicate the shared gene structure of the two MADS genes. We have also revised the legend and added this text to lines 159-162. “The MADS15 and MADS221 genes are predicted to have the same overall gene structure. As targeting constructs had two sgRNAs, two scissors are shown to indicate the approximate cut sites in exon 1.”
2 The authors obtained a total of 44 transformed events which were confirmed by PCR, how to define the “transformed events”, how to do it, there was no results about the PCR.
The details on the event confirmation are found in the methods section as stated in lines 474-481.
“The resulting regenerants were verified via PCR from tissue-culture plantlet leaves (Figure 2D) using the following primers: Malus x domestica elongation factor 1 (MdEF1, F5’-GACATTGCCCTGTGGAAGTT-3’, R 5’-GGTCTGACCATCCTTGGAAA-3’), used to assess competence of DNA for PCR; VirG (F 5′-GCCGGGGCGAGACCATAGG-3′, R 5′-CGCACGCGCAAGGCAACC-3′) used to assess whether there is residual, endophytic Agrobacterium contamination; and Cas9 and nos terminator specific primers to check for transgene presence (Cas9_end_F2: 5’-CCTACAACAAGCACCGGGAT-3’ and Cas9_tnos_R2: 5’-AACGATCGGGGAAATTCGAG-3’).”
We have changed the title of this part of the methods section from “apple transformation” to “apple transformation and event validation.”
3 The results on “transformed events” and “Mutation types and frequency in target genes” were confusing. The authors lack the basic knowledge of plant biotechnology, major revision is needed for the results parts.
No actionable items were provided for the suggested revision.
4 Figure 6, if there was no control, the results were not credible.
As we stated previously, only the single individual tree had reached maturity, so there was no true control. We agree that a comparison with a standard M.26 apple blossom is important. Therefore, we have revised this figure to include a typical flower from the M.26 parent cultivar. Here is the new figure legend.
Figure 6. Floral phenotype of Line T532.
(A) A standard flower from the M.26 parental cultivar showing a typical floral phenotype, including stamens with filaments and anthers, and styles. (B) Line T532 (M.26 parent line) is from the sgRNA22 sgRNA3 construct and has deletions in all four AG-like MADS alleles. It has double petals, stamens with filaments but no anthers, and no pistil or style.
We have added this text the results line 331-335 “By contrast, the M.26 parental cultivar is a single-flowered variety with well-formed stamens. This comparator bloom was photographed from a separate group of trees to provide an example of standard M.26 flowers.”